# End-to-End Video Semantic Segmentation in Adverse Weather using Fusion Blocks and Temporal-Spatial Teacher-Student Learning

**Xin Yang**[1]    **Yan Wending**[2]    **Michael Bi Mi**[2]    **Yuan Yuan**[2]    **Robby T. Tan**[1]

[1]National University of Singapore
[2]Huawei International Pte Ltd

e0674612@u.nus.edu, yan.wending@huawei.com, michaelbimi@yahoo.com,
yuanyuan10@huawei.com, robby.tan@nus.edu.sg

## Abstract

Adverse weather conditions can significantly degrade video frames, leading to erroneous predictions by current video semantic segmentation methods. Furthermore, these methods rely on accurate optical flows, which become unreliable under adverse weather. To address this issue, we introduce the novelty of our approach: the first end-to-end, optical-flow-free, domain-adaptive video semantic segmentation method. This is accomplished by enforcing the model to actively exploit the temporal information from adjacent frames through a fusion block and temporal-spatial teachers. The key idea of our fusion block is to offer the model a way to merge information from consecutive frames by matching and merging relevant pixels from those frames. The basic idea of our temporal-spatial teachers involves two teachers: one dedicated to exploring temporal information from adjacent frames, the other harnesses spatial information from the current frame and assists the temporal teacher. Finally, we apply temporal weather degradation augmentation to consecutive frames to more accurately represent adverse weather degradations. Our method achieves a performance of 25.4% and 33.0% mIoU on the adaptation from VIPER [28] and Synthia [29] to MVSS [18], respectively, representing an improvement of 4.3% and 5.8% mIoU over the existing state-of-the-art method.

## 1 Introduction

Unsupervised domain adaptation (UDA) is gaining attention in video semantic segmentation, offering a solution to the challenge of annotations by adapting models from labeled synthetic datasets to unlabeled real-world scenarios. However, existing video-based UDA methods often falter under the assumption of ideal conditions, neglecting the drastic impact of adverse weather conditions like nighttime and fog. These weather conditions can lead to significant degradation in video quality and result in inaccurate predictions.

The existing UDA methods often rely on two components: pretrained optical flow and pseudo-labels, where the optical flow is used to warp adjacent frames, and the pseudo-labels are used for unsupervised training on the target domain [10, 30, 36, 24, 9]. However, when it comes to adverse weather conditions, the reliability of these components diminishes for two main reasons. Firstly, adverse weather conditions introduce significant degradation in low-level features, including issues such as noise and glare effects during nighttime, as well as occlusions in rainy and foggy conditions. Since existing methods are not inherently designed to handle such low-level degradations, they can

38th Conference on Neural Information Processing Systems (NeurIPS 2024).

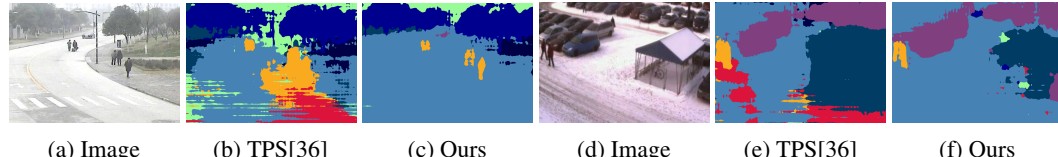

| (a) Image | (b) TPS[36] | (c) Ours | (d) Image | (e) TPS[36] | (f) Ours |

Figure 1: Our model demonstrates enhanced robustness compared to TPS [36] in semantic segmentation tasks under foggy and snowy conditions. It notably excels by significantly reducing inaccuracies in the segmented areas.

easily be misled by these adverse effects, leading to inaccurate predictions [20, 19]. Secondly, as highlighted in [25, 37], adverse weather conditions have distinct styles, which magnify the domain gaps between synthetic datasets and real-world datasets in adverse weather scenarios.

To address these challenges, the novelty of our method lies in introducing the first end-to-end, optical-flow-free video domain adaptation strategy, tailored for real-world videos in adverse weather conditions. Unlike the existing methods, we avoid relying on potentially erroneous optical flows from pretrained models. Instead, we design a fusion block that merges the feature-level information from adjacent frames. We simultaneously train the segmentation model and the fusion block, guided by segmentation losses. Hence, our fusion block learns to combine temporal information which can benefit the semantic segmentation task, from different frames.

We have developed a temporal-spatial teacher-student learning approach to effectively train the fusion block and enhance the quality of pseudo-labels. This approach encompasses two teachers, a temporal teacher and a spatial teacher, who collaboratively instruct a student model. Temporally, the teacher network receives consecutive frames, including the current frame and its adjacent frames. We use the predictions of the current frame as our pseudo-labels. The student network also receives the same adjacent frames, but for the current frame, we provide it with a cropped segment. Then, we enforce consistency between the student network's prediction and the pseudo-label. This compels the student network to actively incorporate temporal information from adjacent frames, enabling it to perform outpainting on the cropped segment and produce the same prediction as the pseudo-label. Spatially, the teacher network benefits from a high-resolution version of the cropped segment to create the pseudo-label, a proven method for enhancing pseudo-label quality, as suggested in [13]. To the best of our knowledge, integrating temporal and spatial modeling using two teachers and one student to achieve an optical-flow-free model is novel. Additionally, the fusion block and its integration into our temporal teacher-student framework have not been explored before.

Augmentation plays a crucial role in enhancing the effectiveness of UDA methods [37, 19, 20]. In the context of adverse weather conditions, certain weather-specific degradations exhibit temporal patterns. For instance, areas with low light in one frame during nighttime are likely to persist in adjacent frames, albeit with potentially varying intensity due to vehicle movement. Similarly, the presence of fog and the accumulation of rain effects also span consecutive frames, with intensity changes influenced by shifts in depth [31, 37]. To effectively capture these characteristics of adverse weather conditions, we introduce a temporal weather degradation augmentation strategy. This strategy involves applying correlated augmentations to either the same or closely positioned locations in consecutive frames, with each undergoing gradual changes in intensity.

Fig. 1 compares our method with TPS [36], illustrating our method's enhanced robustness in adverse weather conditions, achieved independently of pretrained optical flow. In a summary, our contributions are as follows:

- We present an end-to-end, optical-flow-free domain adaptation strategy, by incorporating a fusion block that merges feature-level temporal information. This enables us to bypass the reliance on potentially erroneous optical flows from pretrained models under adverse weather conditions. To the best of our knowledge, this is the first strategy of its kind.

- We introduce a temporal-spatial teacher-student learning method, wherein a temporal teacher guides the student model in gathering information from adjacent frames, and a spatial teacher concentrates on the current frame. These teachers train the fusion block to actively explore the temporal information while effectively harnessing spatial information.

- We develop a temporal augmentation strategy that applying weather degradation augmentations to corresponding or closely positioned locations across consecutive frames. This

approach, featuring gradual intensity variations, effectively captures the dynamic nature of adverse weather degradations.

Our method achieves a performance of 25.4% and 33.0% mIoU on the adaptation from VIPER [28] and Synthia [29] to MVSS [18], respectively, representing an improvement of 4.3% and 5.8% mIoU over the existing state-of-the-art method.

## 2 Related work

**Video semantic segmentation**  Video semantic segmentation aims to label each pixel in video frames while maintaining temporal consistency. Unlike image segmentation, it must address the challenges of temporal coherence and efficiency across sequences. For instance, methods like those in [17, 35, 32, 22] capture temporal information from consecutive frames by leveraging supervision from existing labels.

**Domain adaptive video semantic segmentation**  UDA techniques are extensively applied in various computer vision tasks [33, 31, 8, 23, 12, 13, 37, 20, 2, 4, 1, 39]. Techniques such as adversarial training involving domain discriminators [33, 31] and pseudo-label-based self-learning approach [8, 12, 13, 37, 20] are commonly employed in these methods. The primary function of these approaches is to adapt models from a labeled source domain (for instance, under clear weather conditions) to an unlabeled target domain (like adverse weather conditions). These techniques enable the model to perform impressively in the target domain, despite the absence of ground truth labels.

Recent studies have sought to expand UDA methods from image-based to video-based tasks as a means to circumvent the labor-intensive and costly process of labeling videos [10, 30, 36, 9, 24]. These works typically utilize synthetic datasets like VIPER [28] and Synthia [29] as their source domains, where semantic segmentation ground truths are automatically generated due to their synthetic nature. As for the target dataset, they use a real-world urban scene dataset, Cityscapes-Seq [7]. These studies successfully develop models capable of making predictions on both synthetic and real-world datasets, thus eliminating the need for manual labeling of the real-world data.

Among these methods, DA-VSN [10] employs a temporal domain discriminative loss to minimize the differences between source and target domains and uses an intra-domain consistency loss to improve the accuracy of less confident target predictions. VAT-VST [30] introduces a two-stage UDA method, initially utilizing a sequence domain discriminator to bridge domain gaps, followed by a second stage that employs a pseudo-label-based self-learning approach. This approach aggregates predictions from several preceding frames to create pseudo-labels for the current frame. TPS [36] presents a cross-frame augmentation and pseudo-labeling technique, where predictions from adjacent frames serve as pseudo-labels for the current frame. SFC [9] develops a Segmentation-to-Flow Module (SFM) to involve optical flow in the training of the semantic segmentation model. Random augmentation applied to the current frame are then reconciled with these pseudo-labels through a consistency loss, training the model to become robust to these augmentation. It's important to highlight that all these methods depend on pretrained optical flow estimations: DA-VSN uses it for intra-domain consistency loss, VAT-VST for aggregating predictions, TPS for warping pseudo-labels, and SFC for additional supervision.

**Adverse weather degradation**  Current video-based UDA methods are mainly developed for adapting models from synthetic to real-world datasets under ideal conditions. However, they fall short in adverse weather conditions. This limitation is largely due to two key factors in the domain gap between synthetic and real-world scenes under such conditions: style-related differences, and significant low-level degradations [25, 21, 19, 3, 5].

The style-related gap refers to the stylistic disparities between synthetic and real-world datasets, which UDA typically addresses by training models to recognize both styles. In scenarios like Cityscapes-Seq [7] with ideal conditions, low-level degradations are minimal, allowing existing methods to primarily tackle the style-related gap. But in adverse weather, as [19] discusses, these degradations can severely distort features. For example, a car might be obscured by glare from headlights, leading to erroneous feature extraction. Such challenges render pseudo-labels and pretrained optical flows unreliable. Therefore, our research focuses on overcoming these obstacles by proposing a video-based UDA method specifically designed for adverse weather conditions.

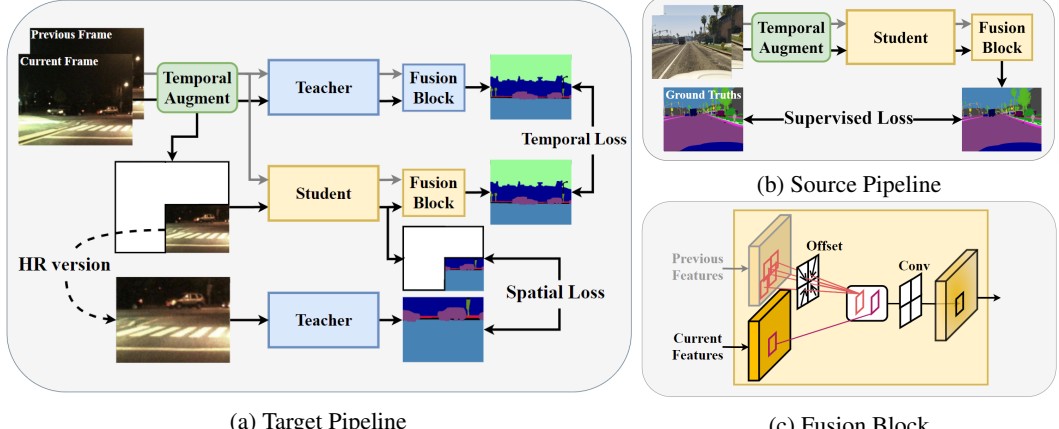

(a) Target Pipeline

(b) Source Pipeline

(c) Fusion Block

Figure 2: Our network comprises two pipelines: the source and the target. (a) Target Pipeline: The upper teacher (temporal) takes both the current and adjacent frames to create temporal pseudo-labels. The student, on the other hand, receives a cropped segment of the current frame and a complete adjacent frame, with a loss function enforcing its predictions align with the temporal teacher. The lower teacher (spatial) uses the same segment as the student, but from the original image and at a higher resolution. Similarly, a consistency loss is applied to make the student's predictions consistent with the spatial teacher's pseudo-labels. (b) Source Pipeline: The student model undergoes supervised learning with consecutive frames as inputs. (c) Fusion Block: This component integrates multiple offset layers, which adjust pixels from adjacent frames relative to the current frame, and convolutional layers to merge these pixels.

## 3 Proposed method

Our proposed method is designed to train a video semantic segmentation model capable of handling adverse weather conditions using an UDA approach. Distinguishing itself from existing methods, ours operates efficiently without the requirement of optical flows. In the source pipeline, we leverage synthetic datasets and their corresponding ground truths for supervised training. This pipeline takes two inputs: the current frame and an adjacent frame. Upon applying temporal weather degradation augmentation to both the current and adjacent frames, they are then input into our network. Subsequently, the network processes the two inputs individually, producing separate sets of feature maps for each frame. These feature maps are then fused by the fusion block, resulting in the final prediction for the current frame. A supervised loss is computed based on the prediction and the ground truth for the current frame,

$$\mathcal{L}_{\text{sup}} = -\frac{1}{N} \sum_{i=1}^{H} \sum_{j=1}^{W} \sum_{c=1}^{C} y_{ijc} \log(p_{ijc}), \tag{1}$$

where, H and W are the height and width of the image, respectively. C is the number of classes. $y_{ijc}$ is from the ground truths indicating whether the class label $c$ is the correct classification for the pixel at position $(i, j)$. $p_{ijc}$ is the predicted probability of the pixel at position $(i, j)$ belonging to class $c$. $N$ is the total number of pixels considered in the calculation.

As for the target pipeline, we use real-world video frames captured from adverse weather conditions, without ground truths. Within this pipeline, we implement a temporal-spatial teacher-student system involving two teacher models: a temporal teacher and a spatial teacher, and a student model. The temporal teacher processes two complete frames and uses its predictions as temporal pseudo-labels. For the student model, we employ the complete adjacent frame and generate a cropped segment from the current frame. Similarity to the source pipeline, we apply temporal weather degradation augmentation to both the cropped segment of the current frame and the complete adjacent frame. These augmented frames are then fed into the student model. A temporal consistency loss ensures that the student's predictions, derived from various augmentations, align with the pseudo-labels provided by the temporal teacher. Therefore, using the cropped segment of the current frame compels the student to actively extract temporal information from the adjacent frame. Meanwhile, the temporal

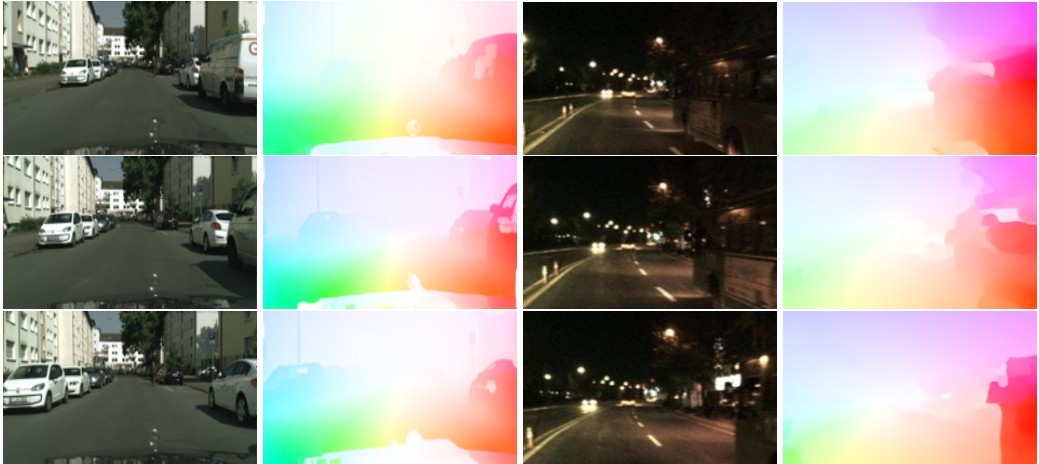

Figure 3: An illustration of optical flows generated using a pretrained FlowNet2 model [27]. The optical flows are generated by utilizing information from the corresponding frame and its previous frame. The left two columns display frames and optical flows under ideal conditions, while the right two columns depict frames and optical flows under adverse weather conditions, with nighttime as an illustrative example. Under ideal conditions, the optical flows accurately capture vehicle details, traffic signs, and poles. In contrast, optical flows under nighttime conditions exhibit significant failures, with missed detection of the middle poles, and erroneous predictions for the bus.

weather degradation augmentation equip the student model to handle real-world conditions where weather-specific degradations often extend across consecutive frames.

Furthermore, the spatial teacher is provided with a high-resolution version of the cropped segment, without any augmentation. The predictions made by this teacher before the fusion block act as spatial pseudo-labels. A spatial consistency loss is then applied, comparing the student model's predictions before the fusion block with the spatial pseudo-labels. This process is designed to direct the student model to effectively harness spatial information from the current frame.

## 3.1 End-to-end training with fusion block

In video-based UDA, where ground truths are unavailable for the target pipeline, existing methods rely on pseudo-labels. A common practice in these methods involves using predictions from the previous frame as pseudo-labels [30, 10, 36, 24, 9]. These pseudo-labels are then warped onto the current frames based on optical flows generated from pretrained models, providing pseudo-labels for the current frame. Subsequently, various techniques are employed to leverage these pseudo-labels for unsupervised training on the current frame [10, 30, 36, 9].

While this approach has shown promise in adapting from synthetic to real-world domains, it faces challenges in adverse weather conditions. Severe weather conditions can significantly distort visual appearance, leading to incorrect pseudo-label generation. The optical flow models, originally pretrained for ideal weather, also suffers from the substantial domain gap between ideal and adverse weather conditions [40, 6].

An example in Fig. 3 highlights this difference. In the example, we utilize the same pretrained optical flow model (FlowNet2) [27] used in existing works. Optical flow predictions under ideal conditions precisely depict object details in images, such as vehicles, poles, and traffic signs, enabling accurate warping of pseudo-labels from adjacent frames to the current frame. Conversely, optical flow predictions under adverse weather conditions, such as nighttime, exhibit the model's inability to identify the movements of distant cars and middle poles, as well as imprecise tracking of the bus.

To overcome this challenge, we propose an end-to-end approach that eliminates the reliance on pretrained optical flow. This training approach comprises a fusion block and a temporal teacher model. The fusion block is specifically designed to merge feature-level information from both the current frame and its adjacent frames, thereby incorporating temporal information for refining predictions on the current frame.

**Fusion block**  The fusion block provides the model an alternative to merge the information from consecutive frames. This is achieved by matching the relevant pixels from adjacent frames, then fusing the matched information. We use deformable convolutional layers as offset layers for matching pixels. We first obtain features from the current frame, denoted as $\mathcal{F}_{\text{cur}}$. The offset layers then map information from adjacent frames to the current frame, resulting in $\mathcal{F}_{\text{adj}}$. Subsequently, both features are concatenated and fused with a convolutional layer to form a new $\mathcal{F}_{\text{cur}}$. This process is repeated several times. Offset and fuse layers are trained end-to-end with segmentation losses, enabling the fusion block to merge beneficial information from adjacent frames for semantic segmentation.

## 3.2  Temporal-Spatial Teacher-Student learning

The teacher-student learning paradigm has been increasingly utilized in image-based UDA [8, 16, 37, 13, 20]. In this approach, the teacher and student models share identical architectures. The teacher model's parameters are updated using the Exponential Moving Average (EMA) of the student model's weights, while the student model is refined through backpropagation with custom loss functions.

Within our proposed methodology, we introduce a dual-teacher system to collaboratively steer the student model. This system comprises a temporal teacher, tasked with enhancing the model's ability to harness temporal information across consecutive frames, and a spatial teacher, focused on extracting and utilizing spatial details from the current frame. It is noteworthy that the temporal teacher's architecture mirrors that of the student model, while the spatial teacher differs by excluding the fusion block. This architectural distinction is clearly illustrated in Fig. 2.

In the temporal dimension, our model is designed to self-sufficiently extract temporal information from consecutive frames, diverging from traditional methods that rely on pre-trained optical flows for information warping. This is achieved by presenting the student model with a randomly cropped rectangular segment comprising 25% of the current frame alongside its fully intact neighboring frames. The locations of the rectangle is selected randomly in each iteration of the training process. Consequently, throughout the entire training process, the model encounters different scenarios where the locations and content of the cropped regions vary. In contrast, the temporal teacher processes the entire current frame. The fusion block then combines the feature maps from two complete frames, utilizing them to generate pseudo-labels. These pseudo-labels further guide the student in compensating for the missing information in the cropped frame segment, following a temporal loss:

$$\mathcal{L}_{\text{temp}} = -\frac{1}{N} \sum_{i=1}^{H} \sum_{j=1}^{W} \sum_{c=1}^{C} y_{ijc}^{\text{temp}} \log(p_{ijc}), \tag{2}$$

where, $y_{ijc}^{\text{temp}}$ is derived from the pseudo-labels. Since the student model must derive the missing information solely from its adjacent frame, the fusion block is specifically trained to harness temporal cues from these frames to reconstruct a complete prediction for the current frame. Consequently, our model demonstrates proficiency in synthesizing temporal information in an end-to-end manner. The entire process is steered by a semantic task-specific loss, ensuring that the fusion block is precisely tailored to this task. It selectively merges information from adjacent frames that is beneficial for semantic segmentation. This targeted fusion, guided by the semantic segmentation loss, distinguishes our approach from existing methods by focusing the training on semantically relevant features rather than indiscriminate information amalgamation.

Spatially, our approach within the target pipeline integrates an established method, as delineated in [13]. We adopt this method to ensure the student model can incorporate information from the current frame with fidelity; it is particularly included to preserve and possibly improve the model's spatial accuracy while it learns to integrate temporal information. For the student model, feature maps are extracted from the cropped segment of the current frame prior to their introduction to the fusion block. Conversely, the spatial teacher is provided with a high-resolution variant of the same cropped segment, from which we also derive feature maps before they reach the fusion block. We apply a spatial consistency loss directly to the feature maps to align the student's learning with that of the spatial teacher:

$$\mathcal{L}_{\text{spat}} = -\frac{1}{N} \sum_{i=1}^{H} \sum_{j=1}^{W} (\mathcal{F}_{ij}^{\text{spat}} - \mathcal{F}_{ij}^{\text{stud}})^2, \tag{3}$$

where, $\mathcal{F}^{\text{spat}}$ are the features from the spatial teacher, and $\mathcal{F}_{ij}^{\text{stud}}$ are the features from the student. $\mathcal{F}^{\text{spat}}$ is resized to match the dimensions of the cropped segment for loss computation. The efficacy

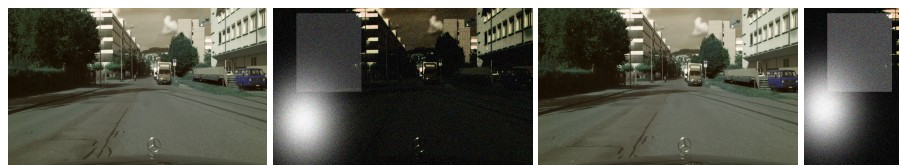

| (a) Previous Frame | (b) Previous with Augs. | (c) Current Frame | (d) Current with Augs. |

Figure 4: This illustration demonstrates the temporal weather degradation augmentation technique. For enhanced visualization, we have utilized Cityscapes-Seq as an example. Frames (a) and (b) are consecutive frames captured from a real-world scene under ideal conditions. Frames (c) and (d) show the same frames, but with applied augmentation, including random noise, a moving glare, a rectangle "foggy" area with intensity change, and a changing illumination.

of this technique for guiding the student model in learning spatial information from unlabeled target images (in the context of our work, the current frames) has been validated in [13]. Thus, this loss safeguards the spatial precision of the model and may also enhance it.

As such, our model innovatively integrates a fusion block, trained using insights from a temporal teacher, to weave together information from consecutive frames without depending on pretrained optical flows. This enables the model to inherently learn and apply temporal details for better current frame predictions. Meanwhile, the spatial teacher ensures the model's spatial accuracy is not compromised and even enhances its capability to extract spatial information.

## 3.3 Temporal weather degradation augmentation

Adverse weather conditions introduce two primary categories of degradation in vision tasks: random disturbances, such as noise and occlusions, and specific weather-related degradations, like low-light, glare, and fog. These degradations occur in similar locations across consecutive frames but exhibit varying intensities due to the movement of objects and the camera.

Our model is strategically developed to counteract such degradations by leveraging the temporal information from consecutive frames. Our objective is to train it to accurately discern the real scene obscured by weather-specific degradations, through a detailed analysis of the variations in their intensity. To achieve this, we simulate weather-induced impairments, including blur, glare, and changes in illumination and chromaticity to both the source images in the source pipeline and the target images used for the student model in the target pipeline. These augmentations are consistently applied to corresponding regions in consecutive frames, with incremental variations in intensity to mimic the dynamic nature of weather-related visual degradations. An illustration of the augmentation are presented in Fig. 4.

Further, a consistency loss is employed to ensure that predictions from the augmented frames align with their corresponding ground truths or pseudo-labels. Hence, these augmentation strategically trains the model to discern the authentic scene behind weather-induced visual distortions by leveraging the variability of degradation intensities across frames.

Overall, our pipelines can be described as follows: Let the input image at frame $t$ be denoted as $X_i$, with the student encoder as $S$ and the teacher encoder as $T$. We define the student fusion block as $F_S$ and the teacher fusion block as $F_T$. Thus, for the temporal pipeline, we impose the following consistency,

$$F_S(S(A_{\text{TWD}}(X_{t-1})), S(Crop(A_{\text{TWD}}(X_t)))) = F_T(T(X_{t-1}, X_t)), \tag{4}$$

where, $A_{\text{TWD}}$ represent the temporal weather degradations, and $Crop$ indicates that the model is provided with only a cropped segment of the current frame. By enforcing this consistency, we encourage the student model to align with the teacher's performance. As a result, the student model learns to be robust against weather degradation while effectively utilizing information from $X_{t-1}$ to compensate for missing details in the cropped current frame.

For the spatial pipeline, we enforce the following,

$$S(Crop(A_{\text{TWD}}(X_t))) = T(\hat{X}_t), \tag{5}$$

Table 1: Quantitative results of our method compared to existing UDA methods, with both image-based and video-based, evaluated against MVSS [18]. **Bold** numbers are the best scores, and underline numbers are the second best scores. The IoU (%) of all classes and the average mIoU (%) are presented. Our method outperforms the best existing method by 4.3 mIoU (%) in average, even with the absence of pretrained optical flows (NOOF).

| Method | Design | car | bus | moto. | bicy. | pers. | light | sign | sky | road | side. | vege. | terr. | buil. | mIoU |
|---|---|---|---|---|---|---|---|---|---|---|---|---|---|---|---|
| \multicolumn{16}{c}{UDA from Synthetic to Real under **adverse** weather condition: VIPER → MVSS} |
| Source-only | Image | 39.4 | 2.6 | 0.0 | 0.0 | 27.3 | 13.6 | 0.6 | 39.4 | 6.6 | 4.2 | 46.2 | 20.2 | 38.2 | 18.3 |
| AdvEnt[33] | Image | 38.3 | 4.8 | 0.5 | 0.0 | 26.3 | **14.6** | 0.7 | 40.7 | 18.2 | 4.4 | 45.5 | 20.7 | 39.1 | 19.5 |
| FDA[38] | Image | 38.5 | 2.2 | 0.3 | 0.5 | 21.8 | 10.7 | 0.8 | 41.8 | 29.4 | 4.5 | 51.4 | 22.5 | 39.7 | 20.3 |
| RDA[15] | Image | 33.3 | 5.9 | **0.9** | 1.0 | 21.9 | 8.2 | 2.1 | 43.1 | 37.3 | **5.1** | 49.9 | 22.8 | 42.6 | 21.1 |
| DA-VSN[10] | Video | 36.0 | 0.6 | 0.2 | 0.0 | 21.1 | 0.6 | 0.9 | 45.5 | 34.4 | 4.0 | 50.2 | 23.4 | 49.0 | 20.4 |
| SFC[9] | Video | 41.2 | 4.0 | 0.0 | 0.0 | 21.3 | 5.6 | 0.7 | 41.4 | 36.2 | 4.5 | 47.2 | 20.4 | 38.8 | 20.1 |
| TPS[36] | Video | 45.8 | 5.1 | 0.0 | 0.3 | 18.9 | 0.0 | 0.0 | 39.6 | 39.7 | 3.0 | 49.8 | 20.8 | 39.1 | 20.2 |
| MoDA[26] | Video | 41.7 | 5.7 | 0.0 | **1.3** | 14.2 | 0.2 | 1.4 | 36.3 | 43.3 | 3.4 | 46.0 | **24.7** | 52.4 | 20.8 |
| Ours | Video, NOOF | **46.0** | **8.6** | 0.0 | 0.5 | **30.9** | 1.1 | **2.3** | **46.4** | **60.2** | 2.7 | **56.4** | 20.7 | **54.3** | **25.4** |

where, $\hat{X}_t$ represents the same cropped image segment at a higher resolution. By enforcing this consistency, we ensure that the student model remains robust to weather degradation while preserving spatial precision.

## 3.4 Overall loss

The overall loss of the network is defined as:

$$\mathcal{L} = \mathcal{L}_{\text{sup}} + \alpha(\mathcal{L}_{\text{temp}} + \mathcal{L}_{\text{spat}}), \tag{6}$$

where, $\mathcal{L}_{\text{sup}}$ denotes the supervised loss used in the source pipeline. The temporal loss and spatial loss in the target pipeline are represented by $\mathcal{L}_{\text{temp}}$ and $\mathcal{L}_{\text{spat}}$, respectively. The parameter $\alpha$, set empirically to $0.1$, ensures that the losses in the target pipeline do not become overly dominant.

## 4 Experiments

In this part of the paper, we undertake an extensive analysis of our video semantic segmentation approach. The evaluation starts with an overview of the datasets utilized, along with a detailed break-down of the models and settings implemented. Subsequently, we delve into a detailed examination of our approach, showcasing its capabilities and robustness against a range of challenging weather conditions through both quantitative metrics and qualitative examples. To conclude, we engage in ablation studies to discern the impact and necessity of the distinct components integral to our method.

**Datasets** For our source datasets in the video semantic segmentation work, we select VIPER [28] and Synthia [29] for their extensive collections of labeled, synthetic urban landscape frames. For our target dataset, we have chosen MVSS [18], which is characterized by its diverse collection of real-world urban scenes captured under various adverse weather conditions. Since VIPER, Synthia, and MVSS have different class protocols, we evaluate only the common classes, following existing UDA methods. We assess target domain segmentation performance using Intersection over Union (IoU%), with higher percentages indicating better performance.

**Baseline models** In our experiments, we compare our method with a range of UDA techniques, encompassing both image-based and video-based approaches. To ensure equitable comparison, we adopt the DeeplabV2 architecture [34] across all methods. The image-based and video-based methods are configured and trained according to their standard settings. For our method, in line with recommendations from [36], we use the same optimization strategy. This includes consistent parameters across all methods, such as the number of epochs, batch sizes, learning rates, and the pretrained backbone, Accel [17].

### 4.1 Quantitative results

As shown in Tabs. 1 2, our models outperform other methods on the real-world dataset under adverse weather, MVSS [18]. Our model surpasses the second best method by 4.3% and 5.8% in mIoU,

Table 2: Quantitative results of our method compared to existing UDA methods, with both image-based and video-based, evaluated against MVSS [18]. **Bold** numbers are the best scores, and underline numbers are the second best scores. The IoU (%) of all classes and the average mIoU (%) are presented. Our method outperforms the best existing method by 5.8 mIoU (%) in average, even with the absence of pretrained optical flows (NOOF).

| UDA from Synthetic to Real under **adverse** weather condition: Synthia → MVSS | | | | | | | | | | | | |
|---|---|---|---|---|---|---|---|---|---|---|---|---|
| Method | Design | car | bicy. | pers. | pole | light | sign | sky | road | side. | vege. | mIoU |
| Source-only | Image | 29.0 | 0.5 | 14.5 | 0.7 | **0.2** | 25.2 | 15.8 | 10.0 | 37.2 | 38.6 | 17.2 |
| AdvEnt[33] | Image | 37.6 | 2.4 | 27.0 | 0.5 | **0.2** | 36.1 | 56.4 | 12.9 | 32.2 | 41.7 | 24.7 |
| FDA[38] | Image | 40.0 | **2.5** | 30.9 | **1.7** | 0.1 | 38.1 | 59.5 | 14.9 | 34.8 | 43.7 | 26.6 |
| RDA[15] | Image | 42.3 | 2.3 | 38.4 | 0.0 | 0.1 | 34.0 | 68.3 | 13.1 | 39.7 | 37.0 | 27.5 |
| DA-VSN[10] | Video | 41.9 | 1.2 | 35.7 | 1.1 | 0.0 | 38.0 | 64.6 | 14.0 | 35.1 | 40.5 | 27.2 |
| SFC[9] | Video | 42.7 | 0.5 | 33.0 | 0.0 | 0.0 | 27.2 | 60.6 | 16.2 | 37.1 | 39.4 | 25.7 |
| TPS[36] | Video | 36.4 | 0.7 | 40.3 | 0.0 | 0.1 | 34.0 | 65.7 | 16.0 | 42.0 | 42.5 | 27.8 |
| MoDA[26] | Video | 35.2 | 0.5 | 23.5 | 0.3 | 0.0 | 41.3 | 64.9 | 15.7 | 41.4 | 47.3 | 27.0 |
| Ours | Video, NOOF | **45.1** | 1.5 | **43.1** | 1.2 | 0.0 | **51.1** | **70.7** | **19.5** | **47.4** | **50.6** | **33.0** |

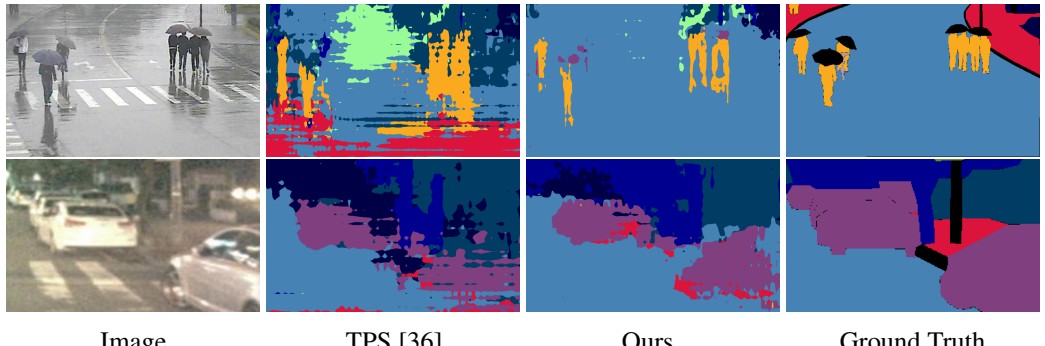

Image        TPS [36]        Ours        Ground Truth

Figure 5: Comparisons on the semantic segmentation performance with TPS [36], Ours, and ground truths on MVSS under rainy and nighttime conditions.

adapting from VIPER [28] and Synthia [29] to MVSS, respectively, our model mark substantial advancements. These consistent gains in IoU across most classes highlights our models' robustness against various adverse weather conditions.

It is worth noting that all video-based methods outperform image-based ones in ideal conditions. However, this advantage does not hold in adverse weather, indicating a failure to effectively use temporal information due to unreliable pseudo-labels and optical flows. In contrast, our end-to-end designed models consistently leverage temporal information under any condition, showcasing their versatility.

## 4.2 Qualitative results

Building on the qualitative insights from Fig. 1 under foggy and snowy conditions, we extend our performance showcase to include rainy and nighttime scenarios in Fig. 5. Our method is evaluated alongside TPS [36] and compared to ground truth segmentation maps. The results highlight that, while TPS tends to yield substantial inaccuracies, our method significantly reduces such errors. This clearly demonstrates our model's robustness in the face of adverse weather conditions.

## 4.3 Ablation studies

We evaluate the effectiveness of each component we implemented on VIPER → MVSS, with the results detailed in Tab. 3. The table reveals that omitting the pretrained optical flow leads to a decrease in performance for the Accel baseline. However, this loss in performance is mitigated once we incorporate our fusion block, underscoring its efficacy as an alternative to pretrained optical flow,

Table 3: Ablation studies of our proposed techniques. We can observe that each component independently contributes to the overall improvement in performance.

| Baselines | | Fus. Blk | Tem. Tea. | Spa. Tea. | Tem. Augs. | mIoU (%) |
|---|---|---|---|---|---|---|
| TPS | Accel (NOOF) | | | | | |
| ✓ | | | | | | 20.2 |
| | ✓ | | | | | 18.6 |
| | ✓ | ✓ | | | | 21.7 |
| | ✓ | ✓ | ✓ | | | 23.8 |
| | ✓ | ✓ | ✓ | ✓ | | 24.3 |
| | ✓ | ✓ | ✓ | ✓ | ✓ | 25.4 |

especially in adverse weather conditions. Furthermore, a gradual improvement in mIoU (%) is evident as more techniques are incorporated, affirming the positive contribution of each component to the overall semantic segmentation performance under adverse weather conditions.

## 5    Conclusion

In conclusion, our novel end-to-end video-based method significantly enhances video semantic segmentation in adverse weather conditions, notably achieving this improvement without the reliance on pretrained optical flows. This method includes a fusion block, a temporal-spatial teacher-student learning system, and a strategy for temporal weather degradation augmentation. Our fusion block effectively merges temporal information from adjacent frames, eliminating the reliance on pretrained optical flows seen in existing works. The teacher-student learning approach uses two teachers: a temporal teacher for guiding the student to explore the temporal information from adjacent frames, and a spatial teacher to train the student to harness spatial information from the current frame. Additionally, we apply temporal weather degradation augmentation to accurately simulate and respond to weather-related degradations in consecutive frames. Upon evaluating our models on MVSS dataset featuring real-world adverse weather conditions, we observed that our approach surpasses many existing image-based and video-based methods in performance.

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

Table 4: Comparison of mIoU and Inference Time for Different Models

| Model | mIoU (%) ↑ | Inference Time (s) ↓ |
|---|---|---|
| FlowNet2+DA-VSN | 20.4 | 0.35 |
| FlowNet2+TPS | 20.2 | 0.17 |
| Ours | **25.4** | **0.11** |

# A  Supplemental material

**Inference time analysis**

Developing efficient video semantic segmentation models has posed a persistent challenge. While numerous accurate architectures exist, their computational demands hinder real-time video frame processing, limiting their usability, particularly in scenarios like autonomous driving under ever-changing conditions.

To address this critical issue, researchers have been exploring time-efficient solutions for video semantic segmentation [17, 11, 14]. For the existing video-based UDA semantic segmentation methods, TPS [36] has made strides by improving the processing speed threefold compared to its predecessor, DA-VSN [10]. Nevertheless, these methods rely on pretrained optical flows, introducing additional time overhead during execution.

In contrast, our proposed approach eliminates the need for this extra step, granting us a notable advantage in terms of execution time. To substantiate this claim, we provide a detailed comparison of inference times in Tab. 4. Inference time is computed by averaging the results from processing 1,000 images on one RTX3090 GPU.

In comparison to existing methods, our approach distinguishes itself by replacing the optical flow generation process with a lightweight fusion block. This substitution not only reduces inference time but also enhances semantic segmentation performance. As a result, our method stands out as a promising candidate for practical deployment in scenarios such as autonomous driving. It excels in streamlining the inference process, aligning with the demand for efficient real-time video semantic segmentation.

**Analysis on ideal conditions**

To demonstrate the generalization ability of our methods, we further assess our approach using Cityscapes-Seq [7], a dataset comprising real-world urban scenes captured under ideal conditions. As shown in Tabs. 5 6, where we adapt the models from VIPER and Synthia, to Cityscapes-Seq [7]. Despite being primarily designed for adverse weather, our models demonstrate effective generalization in ideal conditions, achieving comparable performance to other methods specifically designed for such conditions, even without using the informative optical flow.

**Network configurations**

The detailed network structures of the Fusion Block can be found in Tab. 7. $C$ represents the number of channels, which is defined to be the same as the number of classes. This Fusion Block fuses information from adjacent frames into the prediction of the current frame by matching relevant information from the surrounding pixels of the adjacent frame through the offset layers, and then combining information from different frames. Thus, temporal knowledge is incorporated without the need for optical flows.

Table 5: Quantitative results of our method compared to existing UDA methods, with both image-based and video-based, evaluated against Cityscapes-Seq [7]. **Bold** numbers are the best scores, and underline numbers are the second best scores. The IoU (%) of all classes and the average mIoU (%) are presented.

| Under **ideal** condition: VIPER → Cityscapes-Seq | | |
|---|---|---|
| Method | Design | mIoU |
| Source-only | Image | 37.1 |
| AdvEnt[33] | Image | 44.5 |
| FDA[38] | Image | 44.4 |
| RDA[15] | Image | 44.4 |
| DA-VSN[10] | Video | 47.8 |
| VAT-VST[30] | Video | 48.7 |
| SFC[9] | Video | **51.7** |
| TPS[36] | Video | 48.9 |
| Ours | Video, NOOF | 51.2 |

Table 6: Quantitative results of our method compared to existing UDA methods, with both image-based and video-based, evaluated against Cityscapes-Seq [7]. **Bold** numbers are the best scores, and underline numbers are the second best scores. The IoU (%) of all classes and the average mIoU (%) are presented.

| Under **ideal** condition: Synthia → Cityscapes-Seq | | |
|---|---|---|
| Method | Design | mIoU |
| Source-only | Image | 38.3 |
| AdvEnt[33] | Image | 44.0 |
| FDA[38] | Image | 45.2 |
| RDA[15] | Image | 45.1 |
| DA-VSN[10] | Video | 49.5 |
| VAT-VST[30] | Video | 47.1 |
| SFC[9] | Video | **55.3** |
| TPS[36] | Video | 53.8 |
| Ours | Video, NOOF | 51.0 |


Table 7: Network Structure of Fusion Block

| Fusion Block | |
|---|---|
| **Bottleneck** | Conv, $3 \times 3$, 2C, stride 1, padding 0 |
| **Offset1** | Conv, $3 \times 3$, C, stride 1, padding 1, Sigmoid |
| **Fuse1** | DeformConv, $3 \times 3$, C, stride 1, padding 0 |
| **Offset2** | Conv, $3 \times 3$, C, stride 1, padding 1, Sigmoid |
| **Fuse2** | DeformConv, $3 \times 3$, C, stride 1, padding 0 |
| **Offset3** | Conv, $3 \times 3$, C, stride 1, padding 1, Sigmoid |
| **Fuse3** | DeformConv, $3 \times 3$, C, stride 1, padding 0 |
| **Offset4** | Conv, $3 \times 3$, C, stride 1, padding 1, Sigmoid |
| **Fuse4** | DeformConv, $3 \times 3$, C, stride 1, padding 0 |

The reviewers of your paper will be asked to use the checklist as one of the factors in their evaluation. While "[Yes] " is generally preferable to "[No] ", it is perfectly acceptable to answer "[No] " provided a proper justification is given (e.g., "error bars are not reported because it would be too computationally expensive" or "we were unable to find the license for the dataset we used"). In general, answering "[No] " or "[NA] " is not grounds for rejection. While the questions are phrased in a binary way, we acknowledge that the true answer is often more nuanced, so please just use your best judgment and write a justification to elaborate. All supporting evidence can appear either in the main paper or the supplemental material, provided in appendix. If you answer [Yes] to a question, in the justification please point to the section(s) where related material for the question can be found.

IMPORTANT, please:

- **Delete this instruction block, but keep the section heading "NeurIPS paper checklist",**
- **Keep the checklist subsection headings, questions/answers and guidelines below.**
- **Do not modify the questions and only use the provided macros for your answers**.

