# OpenReview forum: "End-to-End Video Semantic Segmentation in Adverse Weather using Fusion Blocks and Temporal-Spatial Teacher-Student Learning"
_NeurIPS.cc/2024/Conference — NeurIPS 2024 poster_

### Official Review · Reviewer_dY49 · 2024-06-17

**Soundness:** 3
**Presentation:** 3
**Contribution:** 3
**Rating:** 7
**Confidence:** 5

**Summary:**

This paper provides a practical solution for video-based semantic segmentation under adverse weather conditions. Existing methods mainly focus on domain adaptation from synthetic to real data (Viper/Synthia to Cityscapes), but this is the first paper to address videos under adverse weather conditions.

Current approaches heavily depend on optical flows generated from pretrained models to gather temporal information from adjacent frames for pseudo label-based learning. However, this paper highlights that the generated optical flow can be significantly inaccurate due to the degradations caused by adverse weather conditions.
To solve this problem, the authors proposed an end-to-end semantic segmentation model that directly incorporates temporal information from adjacent frames through a temporal and spatial teacher-student learning approach.
To further enhance the model's robustness under adverse weather conditions, the authors introduced a temporal weather degradation augmentation, which mimics real-world scenarios where degradation is similar but varies in intensity across consecutive frames.

The paper conducted experiments on Viper/Synthia to MVSS domain adaptation and achieved state-of-the-art performance even without using optical flow information.

**Strengths:**

1. Video domain adaptation under adverse conditions is an important and practical task.

2. Identifying the issue of optical flows under adverse conditions and proposing a self-learning solution to gather temporal information is novel and contributes to the community.

3. The method achieves good performance even without relying on optical flow information from pretrained models.

**Weaknesses:**

1. According to your method, the spatial loss should be computed between the teacher's output and the corresponding segment of the student's output. This is unclear in Figure 2, as the arrow points to the entire image, including the white area. Is the white area also involved in computing spatial loss?

2. In the caption of Figure 3, it should state, "the left two columns display the frames and optical flows under ideal conditions" and "the right two columns," not "top two rows" or "bottom two rows."

3. For the teacher-student approach:
     - What is the weight smooth coefficient parameter alpha for the EMA updating?
     - How are the pseudo labels selected? Is it done using a threshold?

4. Compared to image-based domain adaptation papers using datasets such as Cityscapes [1], the qualitative results in your paper on MVSS [2] seem poor for both TPS [3] and your method.

5. The inference speed comparison in Table 4 should be highlighted and included in the main paper. Inference speed is crucial for video semantic segmentation, making it a key reason why Accel [4] is preferred over the latest transformer-based methods in this domain. I suggest discussing Accel in the related work section, as it is used in all your video-based benchmarks.

6. Details of the temporal augmentations are missing. For instance, the implementation of "foggy" areas and glare effects should be explained.

7. It would be beneficial to include an overall loss equation, Loverall, that describes how Lsup, Ltemp, and Lspat​ are integrated and weighted.

[1] The cityscapes dataset for semantic urban scene understanding

[2] Multispectral video semantic segmentation: A benchmark dataset and baseline

[3] Domain Adaptive Video Segmentation via Temporal Pseudo Supervision

[4] Accel: A Corrective Fusion Network for Efficient Semantic Segmentation on Video

**Questions:**

Please refer to the weakness.

**Limitations:**

I do not see any potential negative societal impacts.

---

> ### Author Rebuttal · Authors · 2024-08-07
>
> We thank the reviewer for recognizing that our task is important, the idea is novel and contributes to the community, and with a good performance.
>
> __Weakness-1:__ The white area is not involved in computing the spatial loss, we will adjust the arrow to make it correctly pointing to the right-bottom part.
>
> __Weakness-2:__ Thank you for pointing out the issue. Your understanding is correct, and we will revise it accordingly.
>
> __Weakness-3:__ The weight smooth coefficient is 0.9995. Regarding the pseudo-label selection for the teacher-student losses, we use the ratio of pixels exceeding a threshold τ (0.968) of the maximum softmax probability, as suggested in [7].
>
> __Weakness-4:__ Unlike the Cityscapes dataset, which contains images captured under clear conditions using an automotive-grade 22 cm baseline stereo camera and produces high-quality images, the MVSS dataset is more suited to real-world scenarios. The MVSS images are captured from different cameras under various conditions and are of relatively lower quality. The Cityscapes dataset features images with a resolution of 2048 x 1024, whereas the MVSS dataset images have a resolution smaller than 630 x 460. Due to this lower quality, the qualitative results from MVSS inputs do not appear as good as those from Cityscapes.
>
> __Weakness-5:__
> Thank you for the suggestion, we will include Accel in the related work and emphasis the importance of inference speed.
>
> __Weakness-6:__
> For the “foggy” and glare effects, we first randomly select the affected area in the current frame. To create the “foggy” area, we adjust the gamma and chromaticity. The glare effect is produced by using a Gaussian kernel at the selected area, with the center having the highest values that gradually decrease with distance from the center. Once the augmentations in the current frame are completed, we add the augmentations to the adjacent frames. However, the affected area will randomly move around, and the intensity of the augmentations will also vary, mimicking real-world scenarios.
>
> __Weakness-7:__
> Thank you for the suggestion and we will follow accordingly.

---

> > ### Comment · Reviewer_dY49 · 2024-08-08
> >
> > Thank you to the authors for their responses. They have addressed my concerns and I would like to raise my score. I tend to accept this paper.

---

> > > ### Author Response · Authors · 2024-08-10
> > >
> > > Thank you for your positive feedback. We’re pleased that our rebuttal has addressed your concerns, and we sincerely appreciate you raising the score.

---

### Official Review · Reviewer_1Ffg · 2024-06-25

**Soundness:** 3
**Presentation:** 2
**Contribution:** 3
**Rating:** 7
**Confidence:** 5

**Summary:**

This paper proposes a video segmentation method for adverse weather conditions by using the unsupervised domain adaptation paradigm.

Its general idea is to introduce the temporal information from adjacent video frames by the proposed spatial-temporal teacher-student learning scheme.

Notably, the proposed method is end-to-end, and does not rely on optical flows from pretrained models.

In addition, a temporal weather degradation augmentation is introduced to mimic the real-world weather variations across consecutive frames.

Experiments show the state-of-the-art performance of the proposed method.

**Strengths:**

+ The proposed method is the first work for video semantic segmentation under the adverse weather conditions, which can benefit the vision community on a variety of tasks and applications.

+ The proposed method is moderately novel and rationale. Most importantly, it is end-to-end, and does not rely on optical flows from pretrained models.

+ A simple but effective temporal augmentation strategy is proposed to augment the weather degradations among consecutive frames.

+ The proposed method significantly outperforms the state-of-the-art methods under a variety of settings.

**Weaknesses:**

- The presentation of this work may focus too much on the high-level conceptual clarity, which leads to the miss of some important experiment and implementation details. For example:

(1) Line 264, could the authors explain what types of adverse weather conditions are in the MVSS dataset?

(2) The module design and configuration of the teacher-student pipeline is missing. Please provide accordingly.

- The related work in this paper is not very extensive. Some important references are missing. For example:

(1) Please cite and discuss STPL [a]. It is a subsequent work following DA-VSN and TPS.
(2) Some more recent works in 2023-2024 on video segmentation [b] and adverse conditions [c] can be discussed.

[a] Spatio-Temporal Pixel-Level Contrastive Learning-based Source-Free Domain Adaptation for Video Semantic Segmentation. CVPR 2023.

[b] Multispectral video semantic segmentation: A benchmark dataset and baseline. CVPR 2023.

[c] Learning generalized segmentation for foggy-scenes by bi-directional wavelet guidance. AAAI 2024.



- More visual segmentation results should be provided. Currently, only Fig.1 and 5 have several visual results.

- There are multiple presentation issues, especially inconsistency, in this submission. For example:

(1) The title of the submission is not consistent with the title in OpenReview. Please unify it.

(2) Line 78-79, mIoU should be presented in percentage, not number. E.g., 4.3% mIoU.

(3) The caption of Fig.3 is difficult to understand, and need to be simplified.

(4) Table 1, 2, 5 and 6. When reporting the performance of ours, ‘Video,’ should be ‘video’.

(5) Inconsistency between ‘Flow2Net’ and ‘FlowNet2’. Please unify it.

- The figures in this paper can be significantly polished. For example:

(1) Fig.2. The teacher/student net and the fusion block in (c) can be more specified.

(2) Fig.3 is not very informative, as it is not the important results or design. Maybe it can be incorporated into Fig.2.

(3) Fig.4. It would be much better to place one type of augmentation on one image.

**Questions:**

- Q1: Line 264, could the authors explain what types of adverse weather conditions are in the MVSS dataset?

- Q2: The module design and configuration of the teacher-student pipeline is missing. Please provide accordingly.

- Q3: The related work in this paper is not extensive. Discuss some more recent works, such as [a,b,c].

- Q4: More visual segmentation results should be provided.

- Q5: Multiple presentation issues.

- Q6: The figures in this paper can be significantly polished.

**Limitations:**

The authors do not provide a limitation discussion.

---

> ### Author Rebuttal · Authors · 2024-08-07
>
> We thank Reviewer 1Ffg for recognizing that our work can benefit the community, novel, effective, and with a significant performance gain.
>
> __Q1:__ The MVSS dataset consists of a total of 52,735 RGB images, with 3,545 of these images annotated. The dataset includes a variety of adverse conditions such as overexposure, nighttime, rain, fog, and snow.
>
> __Q2:__ In the teacher-student pipeline, the weights of both teachers are updated using the following equation:
> $W_t(i+1) = 0.9995W_t(i) + (1 - 0.9995)W_s(i)$, where $W_t(i)$ represents the weight of the teacher models at iteration $i$, and $W_s(i)$ represents the weight of the student model at iteration $i$. For the weighing factors of the teacher-student losses, we use the ratio of pixels exceeding a threshold $\tau \ (0.968)$ of the maximum softmax probability, as suggested in [7]. For all other configurations, we use the same settings as those in the existing methods described in [5, 27] to ensure a fair comparison.
>
> __Q3__: We appreciate the suggestion. We will expand the related work and discuss all the suggested papers.
>
> __Q4__: We have included a more recent method for more qualitative comparison in the attached rebuttal.pdf.
>
> __Q5:__ Thank you for pointing out the presentation issues in the paper, we will revise all of them accordingly.
>
> __Q6:__ We appreciate the suggestion. We will further improve the presentation of the images.

---

> > ### Comment · Reviewer_1Ffg · 2024-08-13
> > **Re: Rebuttal by Authors**
> >
> > Thanks for the authors for provide the rebuttal.
> >
> > My major concerns, especially Q1 and Q2, have been clarified. I have no remaining reason to oppose its acceptance.
> >
> > I would like to accept this paper. I just hope the authors can take the writing comments (Q3-Q6) into account, so that the writing and presentation of this work can be significantly polished.

---

> > > ### Author Response · Authors · 2024-08-14
> > >
> > > Thank you for your support. We’re pleased that our rebuttal has addressed the concerns, and we sincerely appreciate you raising the score. Following the suggestion, we will incorporate the writing comments (Q3-Q6) in our paper revisions.

---

### Official Review · Reviewer_L8p7 · 2024-07-10

**Soundness:** 2
**Presentation:** 3
**Contribution:** 2
**Rating:** 5
**Confidence:** 4

**Summary:**

In this paper, a end-to-end domain-adaptive video semantic segmentation method without optical flow estimation is proposed to address the problem of video frame quality degradation under adverse weather conditions. The proposed method uses the temporal information of adjacent frames through fusion blocks and spatiotemporal teacher models to enhance the model's robustness to video semantic segmentation under adverse weather conditions. The fusion block combines information by matching and fusing relevant information pixels from adjacent frames. The spatiotemporal teacher model includes a temporal teacher and a spatial teacher to guide the student model from the temporal dimension and the spatial dimension, respectively.

**Strengths:**

1. For the first time, an end-to-end video semantic segmentation method without optical flow estimation is proposed, which is suitable for adverse weather conditions.
2. The model's adaptability to real scenarios is enhanced by simulating dynamic weather degradation in consecutive frames.
3. The article achieves significant performance improvements on multiple datasets, surpassing existing state-of-the-art methods.

**Weaknesses:**

1. The paper does not have any comparison or explanation on the amount of calculation and the number of parameters, which makes me worry about the practicality of the method.
2. There are few baselines selected for visual comparison, which makes it difficult to reflect the effectiveness of the proposed method.
3. The paper may lack in-depth discussion and justification of the theoretical basis of the proposed method.
4. Is there some newer sota method that can be compared? The method in the table doesn't seem to be up to date.
5. The work in this article is carried out under severe weather degradation conditions. Therefore, I think that the article should add a section on image restoration in the related works for discussion.

**Questions:**

Please refer to the weaknesses.

---

> ### Author Rebuttal · Authors · 2024-08-07
>
> We thank Reviewer L8p7 for recognizing the importance and performance of our work. Here is our response to the feedback:
>
> __Weakness-1:__
>
> Thank you for the feedback. Our model has an inference time of 0.11 seconds, which is faster than the state-of-the-art baselines: DA-VSN's 0.35 seconds and TPS's 0.17 seconds. In terms of parameters, our model’s size is 173MB, whereas DA-VSN is 619MB (FlowNet part) + 169MB (segmentation part), and TPS is 619MB (FlowNet part) + 169MB (segmentation part). Note that, although TPS is a faster and more accurate version of DA-VSN, both share the same number of parameters.
>
> __Weakness-2 and Weakness-4:__
>
> Thank you for pointing this out. In the main paper, we selected TPS for visual comparison because it has the second-best quantitative performance. In addition to this, we recently found a CVPR’24 paper (published after the NeurIPS’24 submission deadline), but its code has not yet been released. We implemented it ourselves following the motion training provided by the authors. We will include this CVPR’24 paper and the related discussion stated in this rebuttal in our main paper. Additionally, we will add a more thorough survey of this year’s publications to our main paper.
>
> Here is the qualitative comparison. For IoU, higher values are better.
>
> For Table 1, Viper to MVSS domain adaptation,
>
> |Method|Design|car|bus|moto.|bicy.|pers.|light|sign|sky|road|side.|vege.|terr.|buil.|mIoU|
> |------|------|---|---|-----|-----|-----|-----|----|---|----|-----|-----|-----|-----|----|
> |MoDA [1]|Video|41.7|5.7|0.0|1.3|14.2|0.2|1.4|36.3|43.3|3.4|46.0|24.7|52.4|20.8|
> |Ours|Video|__46.0__|__8.6__|0.0|0.5|__30.9__|__1.1__|__2.3__|__46.4__|__60.2__|2.7|__56.4__|20.7|__54.3__|__25.4__|
>
>
> For Table 2, Synthia to MVSS domain adaptation,
>
> |Method|Design|car|bicy.|pers.|pole|light|sign|sky|road|side.|vege.|mIoU|
> |------|------|---|-----|-----|----|-----|----|---|----|-----|-----|----|
> |MoDA [1]|Video|35.2|0.5|23.5|0.3|0.0|41.3|64.9|15.7|41.4|47.3|27.0|
> |Ours|Video|__45.1__|__1.5__|__43.1__|__1.2__|0.0|__51.1__|__70.7__|__19.5__|__47.4__|__50.6__|__33.0__|
>
> We have also included the qualitative comparison in the attached rebuttal PDF.
>
> Pan et al., “MoDA: Leveraging Motion Priors from Videos for Advancing Unsupervised Domain Adaptation in Semantic Segmentation”, CVPR’24
>
> __Weakness-3:__
>
> We have made every effort to address the reviewer’s concerns regarding the lack of in-depth discussion and theoretical justification, which we will elaborate below. However, with all due respect, we are unsure which specific parts of our method require deeper analysis and further theoretical justification. More detailed feedback on this would be much appreciated.
>
> Existing optical flow methods often fail under adverse weather conditions. To address this issue, we use a temporal teacher to guide the student network in learning from adjacent frames, instead of relying on optical flow. Here is our basic idea. We input the current frame into the temporal teacher, which generates predictions used as pseudo-labels. Concurrently, we mask out part of the current frame and feed it into the student network. We enforce a consistency loss between the student’s predictions and the pseudo-labels from the temporal teacher. This approach encourages the student network to reconstruct the masked-out information by leveraging temporal data from adjacent frames, so that it can align its predictions with the pseudo-labels.
>
> To ensure that the student network does not overlook important spatial information while focusing on temporal data, we incorporate a spatial teacher. This integration helps the student model learn and utilize relevant spatial information as well.
>
> As objects move, their features can appear in different locations across frames, making feature alignment challenging as we don’t have optical flow information. To address this, we use a fusion block with deformable convolutional layers to align features from different frames, followed by standard convolutional layers for merging. This block is trained end-to-end with the temporal teacher, enabling the network to effectively integrate temporal information from various frames and improve the model’s semantic segmentation performance.
>
>
> __Weaknesses-5:__
> Thank you, we will follow the suggestion.

---

> > ### Comment · Reviewer_L8p7 · 2024-08-12
> >
> > Thank you to the authors for the rebuttal, which has addressed some of my concerns. However, I still have a few questions that I hope the authors can clarify.
> >
> > How does the comparison of parameter count, computational cost, and speed look for the suboptimal methods SFC and MoDA?
> >
> > Utilizing deformable convolutions to align features from different frames seems to be a very common practice in video processing.
> >
> > Regarding the structure of the paper, I think the main innovation lies in the authors' primary framework, the sub-components seem to be common. Therefore, I suggest that the authors describe the pipeline in conjunction with mathematical formulas to help readers better understand.

---

> ### Author Response · Authors · 2024-08-12
>
> We sincerely thank the reviewer for the further feedback.
>
>
> __Point 1: Comparisons with SFC and MoDA__
>
> _Parameters_
>
> SFC: 406 MB
>
> MoDA: 185MB (Motion part) + 169MB (Segmentation part)
>
> Ours: 173MB
>
> _Inference Speed_
>
> SFC: 0.38s+0.35s (SFC requires two stages, the first for generating robust optical flows, and the second for segmentation)
>
> MoDA: 0.19s+0.17s (Similar to SFC, MoDA also requires two stages)
>
> Ours: 0.11s
>
> All the models we experimented with were run on a single RTX 3090.
>
>
> __Point 2: Utilizing deformable convolutions__
>
> We agree that deformable convolutions to align features is common. We do not actually claim it is our novelty. Our novelty is integrating our fusion block that use deformable convolution into our temporal-spatial teacher-student framework so that optical-flow-free video-based semantic segmentation can be achieved. The deformable convolutions serve as a tool to bridge information across different frames. We will add this to our paper for clarity.
>
> __Point 3: Describe the pipelines with mathematical formulas__
>
> We agree with the reviewer and will follow the suggection. Essentially, we plan to include the following discussion and would greatly appreciate any further feedback.
>
> Let the input image at frame $i$ be denoted as $X_i$, with the student encoder as $S$ and the teacher encoder as $T$. We define the student fusion block as $F_S$ and the teacher fusion block as $F_T$. For the temporal pipeline, we enforce the following consistency:
>
> $F_S(S(A_{TWD}(X_{i-1})), S(\text{Crop}(A_{TWD}(X_{i}))))=F_T(T(X_{i-1}), T(X_i))$
>
> Here, $A_{TWD}$ represent the temporal weather degradations, and Crop indicates that the model is provided with only a cropped segment of the current frame. By enforcing this consistency, we encourage the student model to align with the teacher’s performance. As a result, the student model learns to be robust against weather degradation while effectively utilizing information from $X_{i-1}$ to compensate for missing details in the cropped current frame.
>
> For the spatial pipeline, we enforce the following:
>
> $S(\text{Crop}(A_{TWD}(X_{i})))=T(\tilde{X}_i)$
>
> where $\tilde{X}_i$ represents the same cropped image segment at a higher resolution. By enforcing this consistency, we ensure that the student model remains robust to weather degradation while preserving spatial precision.
>
> Once again, we appreciate your valuable suggestions. We will incorporate the proposed mathematical equations and other feedback into our paper to enhance clarity for our readers.

---

> > ### Author Response · Authors · 2024-08-14
> >
> > Dear reviewer, as the deadline for our discussion is approaching, we kindly ask if you have any further concerns or feedback. Your insights are invaluable to us, and we would greatly appreciate any additional comments or questions you may have.

---

> > ### Comment · Reviewer_L8p7 · 2024-08-14
> >
> > Thanks to the author for the rebuttal. Most of my questions have been answered, so I am willing to upgrade my rating. I strongly suggest that the author supplement the original article according to the content of the rebuttal. In addition, relevant references should also be added.

---

### Official Review · Reviewer_G5E9 · 2024-07-16

**Soundness:** 2
**Presentation:** 2
**Contribution:** 2
**Rating:** 5
**Confidence:** 5

**Summary:**

This paper studies an important task of video semantic segmentation. Specifically, it focuses on adverse weather scenes and proposes an end-to-end, optical-flow-free, and domain-adaptive algorithm by using fusion blocks and temporal-spatial teachers. Extensive experiments are conducted on VIPER, Synthia and MVSS. This is an interesting paper. However, some issues need to be addressed.

**Strengths:**

1. This paper studies an important task of video semantic segmentation under adverse weather conditions.
2. Good results are obtained. It shows clear improvements over compared methods (SFC and TPS).
3. In general, this paper is easy to follow and the proposed modules are easy to understand.

**Weaknesses:**

1. The fusion module, one of the main contributions, does not have much novelty. In my opinion, it is just concatenation of features.
2. Temporal-Spatial Teacher-Student learning seems not novel.
3. It is interesting that the proposed method does not use optical flow but achieve much better performance than those using optical flows. In my opinion, optical flows provide more relevant information and could help a lot in video segmentation. So are the comparisons between optical-flow-based methods and the proposed one fair? Are there factors which could boost performance used in this paper, but not used in previous methods?
4. The discussions about related works are not enough. This task is very close to video semantic segmentation. However, many methods are not discussed and compared with the proposed one. The use of temporal information is largely explored in video segmentation domain and it is necessary to explain how this paper is "technically" different in terms of using temporal information. Those works include, but not limited to: a. Mask Propagation for Efficient Video Semantic Segmentation; b. Coarse-to-Fine Feature Mining for Video Semantic Segmentation; c. Semantic Segmentation on VSPW Dataset through Masked Video Consistency

**Questions:**

Please address the weaknesses above

**Limitations:**

Yes

---

> ### Author Rebuttal · Authors · 2024-08-07
>
> We thank Reviewer G5E9 for recognizing the importance of our task, our good results, and the clarity of our paper.
>
> __Weakness-1 and Weakness-2__
>
> Thank you for your valuable feedback. We will address both Weakness-1 and Weakness-2 together, as they are closely related. Additionally, the novelty of our fusion block should be considered an integral part of our end-to-end framework for temporal modeling, rather than a standalone module.
>
> The innovation of our Temporal-Spatial Teacher-Student learning approach lies in providing a solution for temporal modeling without the use of optical flow, particularly in scenarios where ground truths for the target domain are unavailable. In our case, since a significant part of the current frame is completely erased, there are no clues left for the student model to learn from. Consequently, the student model must extract temporal information from adjacent frames to fill in the erased parts and produce a consistent prediction with the temporal teacher’s pseudo labels. Unlike existing methods that use optical flow to warp temporal information, our approach enables the student model to learn to gather temporal information through this teacher-student learning mechanism. During the learning process, the spatial teacher ensures the spatial precision of the model and may also enhance it, preventing the network from focusing too much on temporal information and neglecting the spatial information in the current frame.
>
> Since we intend to avoid using optical flows under adverse weather conditions, we need to design an alternative solution to merge temporal information for the student network. Therefore, our fusion block is designed to "offset" and "fuse" information from different frames. Objects in consecutive frames can move to different locations due to their motion. For instance, a car in the bottom-left corner of frame t might move to the top-right corner in frame t+1. Directly concatenating features from such frames can cause misalignment due to large displacements. Existing methods use optical flow to "offset" objects between frames before concatenating features. Instead of using optical flow, we integrate deformable convolutional layers with standard convolutional layers. The deformable convolutional layers learn an "offset" to relocate features, followed by standard convolutional layers to "fuse" the relocated features, as illustrated in Figure 2(c). This combination allows our network to gather temporal information without any pretrained optical flows and enables end-to-end training with the Temporal-Spatial Teacher-Student learning approach under adverse weather conditions.
>
> To the best of our knowledge, integrating temporal and spatial modeling using two teachers and one student to achieve an optical-flow-free model is novel. Additionally, the fusion block and its integration into our temporal teacher-student framework have not been explored before. To avoid any misunderstanding, we will clarify this in our paper
>
>
> __Weakness-3:__
>
> Regarding the fairness of our comparison, the backbone for all the models is the same, using AdvEnt, as stated in the experiment section. Apart from the components discussed in our paper, we did not add any other elements to our experiments.
>
> Under clear weather conditions, accurate optical flow can indeed provide strong prior information for generating better predictions. However, under adverse weather conditions, existing optical flow methods can be erroneous, leading to degraded performance. Existing methods use optical flows generated from pretrained FlowNet 2.0 [20], which is trained on clear synthetic datasets and thus become erroneous when applied to adverse weather scenes. Although we could augment clear data with synthetic adverse weather conditions, it is well known that there are significant gaps between synthetic and real weather conditions. Figure 3 in the main paper shows that under clear daytime conditions (left columns), the optical flows are accurate and useful for semantic segmentation. However, under adverse weather conditions (right columns), the optical flow predictions are erroneous. Consequently, using generated optical flows can introduce incorrect information and decrease the performance of existing models.
>
> __Weakness-4:__
>
> Following the suggestion, we will expand the related work section to include a video semantic segmentation subsection, covering [a-c] and other relevant works.
>
> Our use of temporal information differs from that in methods [a-c], which are conventional video semantic segmentation tasks trained with ground-truth labels from datasets like VSPW. In those methods, models learn to capture temporal information from consecutive frames using supervision from these labels. In contrast, our paper focuses on self-supervised domain adaptive video semantic segmentation, which does not rely on ground-truth labels.
>
> Without ground-truth supervision, domain adaptive methods cannot directly learn temporal information. As discussed in our paper, existing methods [4, 5, 18, 23, 27] use optical flow to warp predictions from consecutive frames to generate pseudo labels, thereby “forcing” the model to learn temporal information. However, generated optical flow can be erroneous under adverse weather conditions, as evidenced in Figure 3. Therefore, we develop an optical-flow-free approach to compel the model to learn temporal information.

---

> > ### Comment · Reviewer_G5E9 · 2024-08-08
> >
> > I appreciate authors' responses to my questions.
> >
> > Their answers make sense to me. Since authors promised to make changes in their final version to address weakness 1-4, I tend to accept this paper.

---

> > > ### Author Response · Authors · 2024-08-10
> > >
> > > Thank you for your positive feedback. We're pleased that our rebuttal has addressed your concerns. If you have any further questions or need additional clarification, please let us know.

---

### Author Rebuttal · Authors · 2024-08-07

We thank all the reviewers for their insightful feedback. We are encouraged by their recognition of the importance of our task (G5E9, L8p7, dY49) and its potential benefit to the community (1Ffg, dY49). We are also pleased that they acknowledged the novelty of our method (1Ffg, dY49). Additionally, we appreciate that all the reviewers recognized the effectiveness of our method and noted its clear improvement compared to existing methods.


We have recently found a CVPR’24 paper, MoDA (published after the NeurIPS’24 submission deadline). We include it here for further comparison and will also incorporate it into the main paper.

Below are the quantitive comparisons. For IoU, higher values are better.

For Table 1, Viper to MVSS domain adaptation,

|Method|Design|car|bus|moto.|bicy.|pers.|light|sign|sky|road|side.|vege.|terr.|buil.|mIoU|
|------|------|---|---|-----|-----|-----|-----|----|---|----|-----|-----|-----|-----|----|
|MoDA [1]|Video|41.7|5.7|0.0|1.3|14.2|0.2|1.4|36.3|43.3|3.4|46.0|24.7|52.4|20.8|
|Ours|Video|__46.0__|__8.6__|0.0|0.5|__30.9__|__1.1__|__2.3__|__46.4__|__60.2__|2.7|__56.4__|20.7|__54.3__|__25.4__|


For Table 2, Synthia to MVSS domain adaptation,

|Method|Design|car|bicy.|pers.|pole|light|sign|sky|road|side.|vege.|mIoU|
|------|------|---|-----|-----|----|-----|----|---|----|-----|-----|----|
|MoDA [1]|Video|35.2|0.5|23.5|0.3|0.0|41.3|64.9|15.7|41.4|47.3|27.0|
|Ours|Video|__45.1__|__1.5__|__43.1__|__1.2__|0.0|__51.1__|__70.7__|__19.5__|__47.4__|__50.6__|__33.0__|

We have also included the qualitative comparison in the attached rebuttal PDF.

Pan et al., “MoDA: Leveraging Motion Priors from Videos for Advancing Unsupervised Domain Adaptation in Semantic Segmentation”, CVPR’24

---

### Decision · Program_Chairs · 2024-09-25

**Decision:**

Accept (poster)

**Comment:**

All reviewers highlight the strong contributions. The rebuttal has alleviated the remaining concerns, and all reviewers vote for accepting the paper.